# Gallic acid inhibits EMCV infection via targeting the interaction between TBK1 and IRF3 to promote IFN-β expression

Yan Zhang,[1,2] Yanqiao Wen,[3,4] Weijiao Xue,[1] Tengyu Zhang,[1] Jixia Hou,[1] Xuewen Chen,[3] Ruofei Feng,[3] Chunxia Tan[1,2]

**ABSTRACT** Encephalomyocarditis virus (EMCV) is an important zoonotic pathogen with global distribution, which has caused huge serious economic losses to the development of the livestock industry. Gallic acid (GA) is an active ingredient of traditional Chinese medicine with wide pharmacological and biological activities and is also a potential antiviral drug. However, its antiviral effect and mechanism of anti-EMCV are still unclear. In the present study, we found that therapeutic administration of GA could significantly reduce the viral load and viral titer of EMCV to inhibit EMCV replication in a dose-dependent manner. GA could also protect cells infected by EMCV and decrease the content of capsid protein VP1 of EMCV in HEK-293 cells. Additionally, the mechanistic investigations revealed that GA might exert antiviral effects by regulating the interaction between TBK1 and IRF3 to promote the IFN-β expression. Meanwhile, GA could also alleviate EMCV-infected mice. These results indicate that GA may serve as a novel antiviral agent against EMCV infection.

**IMPORTANCE** As a zoonotic pathogen, encephalomyocarditis virus (EMCV) causes myocarditis, encephalitis, neurological disease, reproductive disorders, and diabetes in pigs, which seriously endangers the development of the swine industry worldwide. However, due to the lack of effective commercial vaccines, there is an urgent need to develop safe and effective drugs against EMCV. Gallic acid (GA) has wide pharmacological and biological activities. However, the antiviral effect and the mechanism of GA are currently unknown. Here, we demonstrated that GA had a significant anti-EMCV effect. Further research found that GA inhibited EMCV infection via targeting the interaction between TBK1 and IRF3 to promote the IFN-β expression. These findings indicate that GA could be an effective anti-EMCV drug.

**KEYWORDS** gallic acid, EMCV, antiviral, IFN-β

Encephalomyocarditis virus (EMCV) is an important zoonotic virus (1), which is widely distributed and can be isolated from various mammals and patients with meningitis. Pigs, being susceptible animals to EMCV, mainly exhibit symptoms such as abortion in the late stage of pregnancy, stillbirth, mummification of piglets in the womb, increased mortality of weak and newborn piglets (2), as well as respiratory diseases in piglets (3). Therefore, EMCV can not only cause encephalitis and myocarditis but also lead to reproductive disorders in sows (4). The pathogenicity of EMCV isolates varies across different countries and regions, and even the same strain exhibits differing pathogenicity toward porcine fetuses of different gestational ages (5). In addition, in severe cases of human infection with EMCV, the clinical manifestations include neurological symptoms such as disturbance of consciousness, convulsions and coma, as well as cardiovascular symptoms such as heart failure and arrhythmia (6, 7). Therefore, EMCV not only has a huge impact on the livestock industry but also seriously endangers human health.

**Peer Reviewer** Huimin Liu, Henan Agricultural University, Zhengzhou, Henan, China

Address correspondence to Ruofei Feng, fengruofei@xbmu.edu.cn, or Chunxia Tan, tancx@sina.com.

Yan Zhang and Yanqiao Wen contributed equally to this article. Author order was determined by drawing straws.

The authors declare no conflict of interest.

Thus, there is an urgent need to develop new drugs for the treatment of EMCV-related diseases.

During a viral infection, the innate immune response is crucial for restricting viral replication and initiating an immune response (8). Interferons (IFNs) can induce host cells to produce antiviral proteins, thereby inhibiting the replication and spread of viruses, which play a particularly important role in the innate immune response to viral infections and provide a powerful first line of defense against invading pathogens. According to sequence homology, IFNs are classified into three families: Type I, Type II, and Type III. Among them, the signal transduction of Type I IFN (IFN-I), also known as "viral interferon," is the core of inducing antiviral innate immunity. IFN-I includes IFN-α, IFN-β, IFN-ε, IFN-κ, and IFN-ω. Currently, IFN-α and IFN-β have been proven to possess antiviral effects and immunomodulatory activities (9), and they can exhibit significant antiviral effects against a wide variety of viruses. The retinoic acid-inducible gene I (RIG-I)-like receptors (RLRs) can recognize viral RNA in the cytoplasm and initiate an immune response by inducing IFN-β. The binding of IFN-β to its receptor induces the activation of JAK1, JAK2, and tyrosine kinase 2, which can phosphorylate of STAT1 and STAT2 to form interferon-stimulated gene (ISG) factor 3 complex (ISGF3). The ISGF3 enters the nucleus and binds to the IFN-stimulated response element, thereby activating the ISG, which induces the antiviral response of the host cell against viral infections (10–13).

Traditional Chinese medicine has unique advantages in antiviral treatment owing to its broad spectrum of actions and targets, along with low side effects. *Gallic acid* (GA) (Fig. 1A), an organic acid, exists extensively in plants such as *Galla Chinensis*, *Camellia sinensis*, and *Hevea brasiliensis*. It has the highest contents in *Galla Chinensis*. Additionally, GA exhibits a variety of biological activities (14, 15). It has been reported that GA could inhibit the activation of the NLRP3 inflammasome and pyroptosis by enhancing the Nrf2 signaling pathway, thereby alleviating rheumatic arthritis (15). It can also reduce lipopolysaccharide-induced renal injury in rats by inhibiting cell pro-death and inflammatory responses (16) and exert anti-tumor effects through multiple biological pathways (17–19). However, currently, there are relatively few studies on the antiviral mechanism of GA. In this study, the inhibitory effect of GA on EMCV was explored, and its anti-EMCV mechanism was further investigated. The study aims to confirm the application prospect of GA against EMCV and provide a theoretical basis for the clinical use of GA in clinical practice and the research of related target drugs.

## MATERIALS AND METHODS

### Cells, viral strains, bacterial strains, and plasmids

The BHK-21 cells (hamster kidney fibroblast cells), HEK293 cells (human embryonic kidney cells), A549 (human non-small-cell lung cancer cells), and EMCV (PV21) virus were all preserved and provided by the Biomedical Research Center of Northwest Minzu University. The cells were cultured at a constant temperature in an incubator under conditions of 37℃ and 5% $CO_2$. The culture media were Dulbecco's modified Eagle medium (DMEM) supplemented with 10% newborn bovine serum (NBS) or F12 medium supplemented with 15% NBS. During the cell resuscitation, culture, and passage, all operations must be carried out strictly in accordance with the aseptic principle. The competent cells of *Escherichia coli* BL21 were preserved and provided by the Biomedical Research Center of Northwest Minzu University. Empty vectors pCMV-HA and pCMV-Flag were provided by the Biomedical Research Center, Northwest Minzu University. The recombinant plasmids pCMV-HA-MDA5, pCMV-HA-TBK1, and pCMV-Flag-IRF3 were all constructed in-house.

### Animals

Male BALB/c mice, 6–8 weeks old, with body weights maintained in the range of 18–22 g, were housed in the Experimental Animal House of the Biomedical Research Center

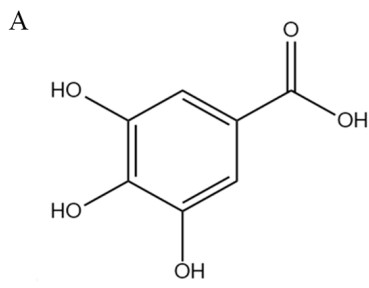

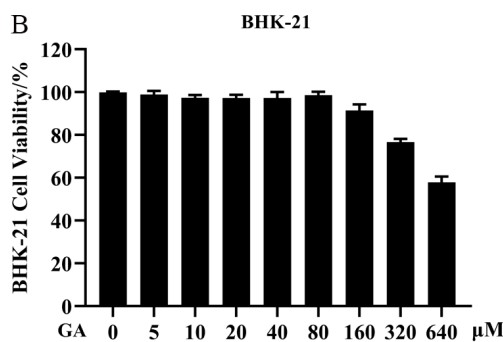

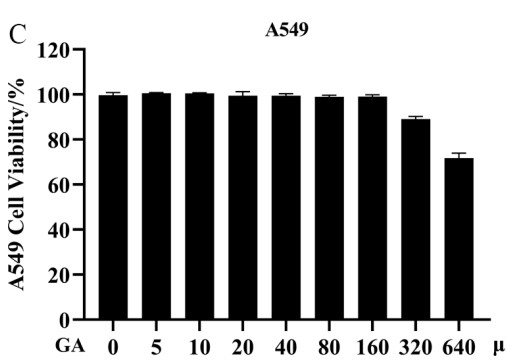

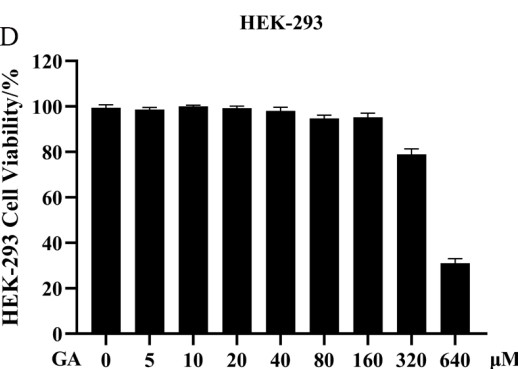

**FIG 1** Chemical structure and cytotoxicity of GA. (A) Chemical structure of GA. Cytotoxicity of GA in BHK-21 cells (B), in A549 cells (C), and in HEK-293 cells (D).

at the Northwest Minzu University. The breeding environment was controlled with a 12-h light-dark cycle, a temperature of 23℃, and a humidity of 40%–60%. Animal welfare and experimental procedures were performed in strict accordance with the Guidelines for the Management and Use of Laboratory Animals (Ministry of Science and Technology of China, 2006) and approved by the Animal Ethics Committee of Northwest Minzu University and the Animal Protection and Utilization Committee (NO. XBMU-SM-2024105).

## Experimental drugs and reagents

GA (149-91-7) was purchased from Shanghai Yuanye (Shanghai, China). NBS was purchased from Lanzhou Minhai (Lanzhou, China). DMEM and 0.25% trypsin were obtained from Lanzhou Bailing (Lanzhou, China). TIFIH1/MDA5 polyclonal antibody (21775-1-AP), MAVS polyclonal antibody (14341-1-AP), and IRF3 polyclonal antibody (11312-1-AP) were purchased from Proteintech (Wuhan, China). Anti-Flag tag rabbit polyclonal antibody (D110005), rabbit polyclonal antibody against the HA tag (D110004), HRP-conjugated goat anti-rabbit IgG (D110058), and HRP-conjugated goat anti-mouse IgG (D110087) were purchased from Sangon Biotech (Shanghai, China). TBK1 antibody (3013S) and phospho-IRF3 (Ser386) (37829 S) were bought from Cell Signaling Technology. GAPDH was purchased from Beyotime Biotechnology (Shanghai, China). Lipofectamine 2000 was purchased from Invitrogen (Carlsbad, CA, USA). IRF3 siRNAs were designed and synthesized by Ribo Biotechnology (Guangzhou, China).

## Experimental methods

### Cytotoxicity experiment

BHK-21, A549, and HEK-293 cells were seeded in 96-well plates and incubated for 24 h. Three groups were established: blank group, control group, and treatment group. In

the blank group, 10% NBS-DMEM (without cells) was added. In the control group, 10% NBS-DMEM (with cells) was added, and 10% NBS-DMEM (with cells) containing different concentrations of GA (5, 10, 20, 40, 80, 160, 320, and 640 µM) was added, with 200 µL per well. Each group had six replicate wells. After 24-h incubation, 10 µL of cell counting kit-8 (CCK-8) solution was added to each well. After further incubation at 37°C for 1.5 h, the absorbance was measured at 490 nm using a microplate reader, and the cell viability was calculated.

### Time addition assay

To assess the impact of GA on different stages of the EMCV life cycle, an addition-time assay was performed. The cells were treated with GA (20–160 µM) at different time points during the EMCV PC21 (100 $TCID_{50}$) infection, including pre-treatment, co-treatment, and post-treatment. The experiment progress is shown in Fig. 2A. The cells and supernatants were collected at 24 h. Subsequently, the samples were subjected to freeze-thaw cycles for three times and centrifuged. The supernatant was carefully harvested, and the total viral genome was extracted using the Tiangen TIANamp DNA/RNA Kit. The extracted RNA was used as a template for reverse transcription into cDNA. Finally, qPCR was used to determine copies of viral genomes.

### Virus titer

Samples of GA at concentrations of 80 and 160 µM, with different treatment times (12, 24, and 36 h), were prepared according to the treatment method described above. After undergoing repeated freezing and thawing cycles, the supernatant was collected. BHK-21 cells were infected with virus samples that had been serially diluted 10-fold in a 96-well plate, with eight replicates for each dilution gradient. Two hours later, the medium was replaced with a maintenance medium consisting of 3% NBS-DMEM. After 3–5 days, the number of cytopathic-positive wells was observed and counted under a microscope, and the virus titer was calculated using the Reed-Muench method.

### Cytoprotective effect

HEK-293 cells were seeded in 6-well plates. When the cells grew to more than 90%, they were infected with EMCV (100 $TCID_{50}$) for 2 h. Subsequently, the culture medium was replaced with 3% NBS-DMEM supplemented with GA (80/160 µM), and the cells were cultured for 24 h. The cell status was observed under a microscope and photographed for documentation.

### RT-qPCR

Total cellular RNA was extracted using the total cellular RNA extraction reagent TranzolUp (TransGen Biotech, Beijing, China). The total RNA was quantitatively reverse-transcribed into cDNA. Then, a two-step RT-qPCR was carried out using a SYBR Green assay with an Applied Biosystems Master Mix kit in an ABI 7500 Real-Time PCR system. GAPDH was used as the internal reference. The relative gene expression levels were calculated using the $2^{-\Delta\Delta Ct}$ method, and the results were presented as the mean value ± standard deviation (SD). Table 1 shows the primer sequences used for RT-qPCR analysis.

### Western blot detection

The HEK-293 cells were lysed on ice for 30 min with a high-efficiency RIPA tissue/cell lysis buffer (Solarbio, Beijing, China). The samples were then subjected to separation via SDS-PAGE and subsequently transferred onto a PVDF membrane (Millipore Corp, Bedford, MA, United States). Next, the membrane was incubated in a blocking buffer (TBST supplemented with 2.5% skimmed milk powder) at room temperature for a duration of 2 h. After incubation, the blocking solution was discarded, and the corresponding primary antibodies were individually added to the membrane, followed by an overnight incubation at 4°C. Subsequently, the corresponding secondary antibodies

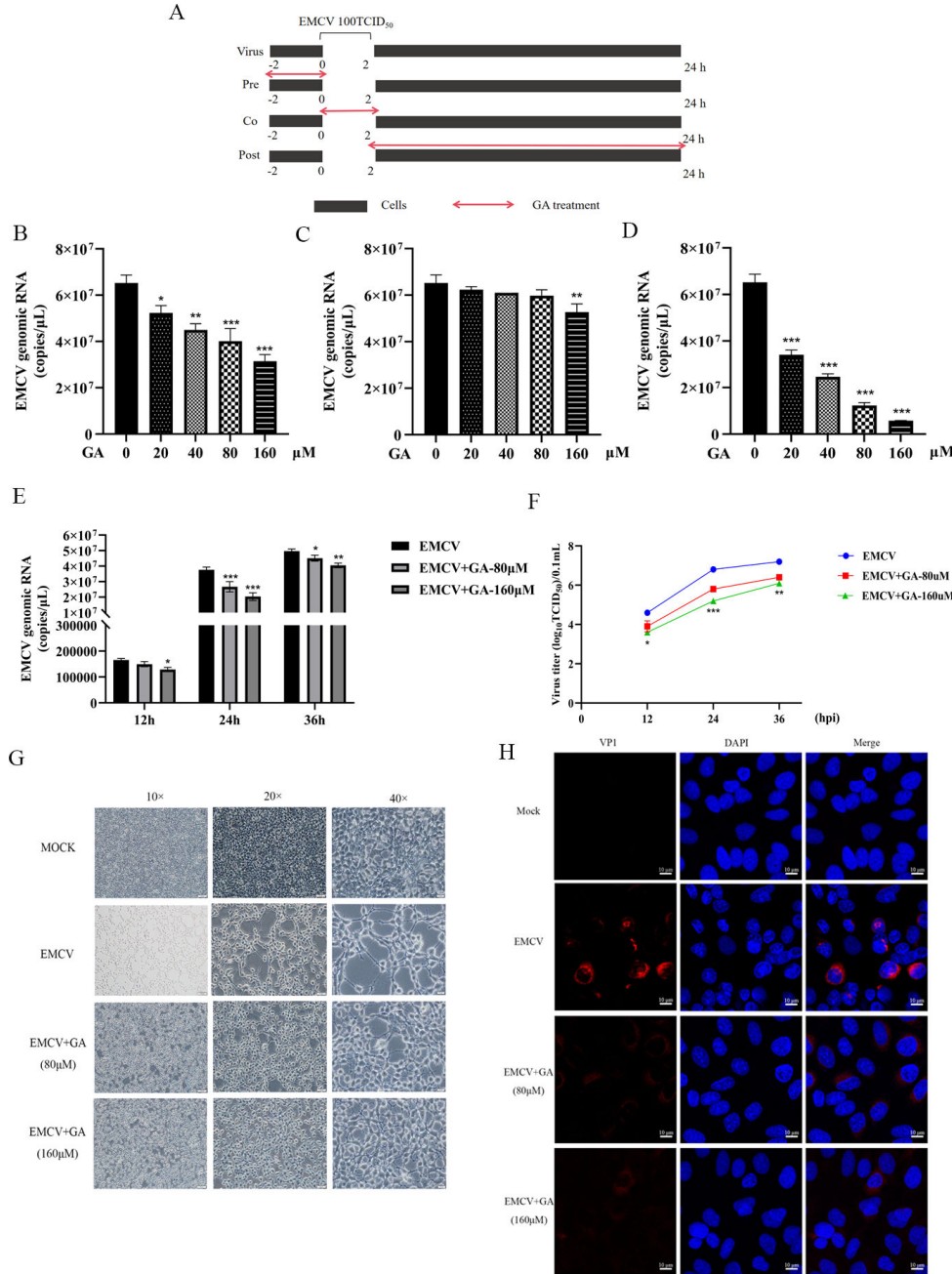

**FIG 2** Antiviral activity of GA against EMCV *in vitro*. (A) Schematic diagram of the time addition experiment. The effects of pre-treatment with GA (B), co-treatment with (C), and post-treatment with (D) on the copies of viral genomes of EMCV in HEK-293 cells. The influence of GA (80/160 µM) on the copies of viral genomes (E) and the virus titer (F) of EMCV (12, 24, and 36 h). (G) The protective effect of GA (80/160 µM) on cells. (H) Immunofluorescence was used to detect the inhibitory effect of GA on the viral protein VP1 of the EMCV. Data were presented as the mean ± standard deviation (SD) of three independent experiments, and *P* values indicate the results of one-way analysis of variance (ANOVA) with Tukey's multiple comparison test or two-tailed t tests. ***P < 0.001, **P < 0.01, and *P < 0.05, compared with the cells not treated with GA.

were incubated with the membrane at room temperature for 1 h. After discarding the secondary antibodies, the PVDF membrane was washed and imaged using ECL (Bio-Rad, CA, United States) chemiluminescence. The gray values of protein bands were analyzed using ImageJ software.

## Plasmid transfection

HEK-293 cells were inoculated into a 6-well cell culture plate. When the cells grew to more than 80%, the plasmid was transfected into the HEK-293 cells via the liposome-mediated transfection approach, which the plasmid HA-TBK1 and Flag-IRF3 were undergone co-transfection during the co-immunoprecipitation assay, while all the others transfected single plasmid. Four to six hours after transfection, the culture medium was refreshed. Following this, the cells were incubated at 37°C for a period of 24 h. Subsequently, the medium was substituted with 10% NBS-DMEM, either supplemented or not supplemented with GA. After an additional 24-h culture period, detection was carried out using RT-qPCR and Western blot analysis.

## RNAi assay

siRNA targeting IRF3 was transfected into HEK-293 cells to verify the effect of IRF3 knockdown. Lipofectamine 2000 was used to transfect HEK-293 cells with siNC or siIRF3 for 24 h, followed by inoculation with 100 $TCID_{50}$ EMCV for 2 h. Subsequently, the culture medium was replaced with 3% NBS-DMEM supplemented with or without GA (160 µM), and the cells were cultured for another 24 h. The viral titers and the expression of IFN-β were tested using the $TCID_{50}$ assay and enzyme-linked immunosorbent assay (ELISA), respectively.

## Immunofluorescence

A549 cells cultured in a 12-well cell culture plate were infected with EMCV (100 $TCID_{50}$) for 2 h. Subsequently, the culture medium was replaced with 3% NBS-F12 medium with or without GA, and the cells were further cultured for 24 h. After that, the cells were fixed with 4% paraformaldehyde, permeabilized with 0.1% Triton X-100, and blocked with a blocking buffer for 30 min. Later, the cells were incubated with the VP1 primary antibody (1:1,000) overnight at 4°C, followed by incubation with Cy3-conjugated goat anti-mouse IgG (diluted 1:500) at 4°C for 2 h and then stained with 4′,6-diamino-2-phenylindole (DAPI) for 10 min. Finally, images were visualized on a ZEISS LSM900 laser confocal microscope (Carl Zeiss, Oberkochen, Germany).

## Co-Immunoprecipitation

HEK-293 cells were lysed with lysate on ice for 30 min to extract total protein. A portion of the protein samples was directly subjected to Western blotting assays to verify the presence of the target protein. The tag antibody was incubated with magnetic beads on a shaker at 4°C for a duration ranging from 4 to 6 h. Subsequently, the remaining untreated protein samples were added, and the mixture was incubated on a shaker at 4°C overnight. The supernatant was carefully discarded, while the magnetic beads at

**TABLE 1** Primer sequences for RT-qPCR

| Primers | Sequences (5′ to 3′) |
| --- | --- |
| Homo-IFN-β-qF | TGCTCTGGCACAACAGGTAG |
| Homo-IFN-β-qR | CAGGAGAGCAATTTGGAGGA |
| Homo-TNF-α-qF | CTGGGCAGGTCTACTTTGGG |
| Homo-TNF-α-qR | CTGGAGGCCCCAGTTTGAAT |
| Homo-IL-6-qF | AACCTGAACCTTCCAAAGATGG |
| Homo-IL-6-qR | TCTGGCTTGTTCCTCACTACT |
| Homo-Bax-qF | TCAGGATGCGTCCACCAAGAAG |
| Homo-Bax-qR | TGTGTCCACGGCGGCAATCATC |
| Homo-Bcl-2-qF | ATCGCCCTGTGGATGACTGAGT |
| Homo-Bcl-2-qR | GCCAGGAGAAATCAAACAGAGGC |
| Homo-GAPDH-qF | GTCTCCTCTGACTTCAACAGCG |
| Homo-GAPDH-qR | ACCACCCTGTTGCTGTAGCCAA |

the bottom layer were retained. The magnetic beads were then washed five times with pre-cooled PBS. The precipitates were combined with SDS buffer and boiled at 95°C for 5 min. Finally, the complexes were analyzed using Western blotting.

## Animal experiments

### Acute toxicity of GA in mice

The initial dose of GA was set at 1,800 mg/kg and then successively reduced in a 3/4 concentration gradient. A total of five concentration gradients were established. The mice were administered different concentrations of GA via gavage at a dosage of 0.2 mL/10 g body weight, and the mortality of the mice was observed. The death status of the mice at each concentration gradient was recorded to calculate the median 50% lethal dose ($LD_{50}$) of GA.

### Establishment of an EMCV-infected mouse model

The experimental design is shown in Fig. 6A. The BABL/c mice were randomly divided into the mock group, EMCV group, L-GA (40 mg/kg) group, and H-GA (80 mg/kg) group with 13 mice in each group. On the fifth day, when deaths occurred in the EMCV group, five mice were randomly sacrificed from each group for other tests such as copies of viral genomes determination, and other eight mice were used for plotting the survival curve. Except for the mock group, the mice in other groups were intramuscularly injected with 250 μL/10 g of 100 $TCID_{50}$ EMCV. Post-infection, the mice were gavaged daily with the corresponding dose of GA with a treatment volume of 0.2 mL/10 g and the corresponding volume of PBS solution in the mock and EMCV groups. The body weight was monitored, and the clinical score was recorded daily. The clinical symptom scoring criteria are as follows: 0: healthy; 1: the mouse is listless; 2: loss of appetite; 3: head edema and ruffled fur; 4: hind-limb paralysis and neurological symptoms; 5: death. The remaining mice were continuously cultured for 3 days after all the mice in the EMCV-infected group had died, and a survival curve was plotted.

### Enzyme-linked immunosorbent assay

The contents of IFN-β in cell supernatants and brain tissues of mice were detected using a human and mouse IFN-β ELISA kit (Multi Sciences, Hangzhou, Zhejiang, China) according to the manufacturer's instructions.

## Statistical analysis

Statistical analysis was performed with SPSS version 26.0 (IBM, Armonk, NY, United States). All data were presented as the mean ± standard deviation (SD) of at least three independent experiments unless otherwise noted. All the figures were plotted using GraphPad Prism software (Version 8.0, La Jolla, CA, United States). Statistical significance was determined via one-way ANOVA with Tukey's multiple comparison test or two-tailed t tests. Statistical significance was set at $P < 0.05$.

## RESULTS

### Cytotoxicity of GA

First, the cytotoxicity of GA in BHK-21, A549, and HEK-293 cells was investigated using the CCK-8 assay. The measurement results indicated that GA exerted cytotoxic effects at a concentration of 640 μM in A549 and HEK-293 cells, and cytotoxicity was detected at a concentration of 320 μM in BHK-21 cells (Fig. 1B through D). Consequently, all subsequent experiments were conducted within the safe concentration range (5–160 μM) of GA.

## GA significantly inhibits the proliferation of EMCV

To analyze the time of GA exerted antiviral effects, GA was applied at different time points for intervention. It was found that compared with the infected group, during the pre-treatment stage before EMCV infection, GA could effectively inhibit the copies of viral genomes of EMCV (Fig. 2B). When GA was applied simultaneously with EMCV infection (co-treatment), GA had no significant antiviral effect on EMCV (Fig. 2C). GA showed the best inhibitory effect during the post-treatment after EMCV infection, and the concentrations ranging from 20 to 160 µM of GA could effectively inhibit the replication of EMCV (Fig. 2D). Meanwhile, we further detected the copies of viral genomes of EMCV and viral titer to confirm the antiviral effect of GA at different time points during EMCV replication using RT-qPCR and $TCID_{50}$ assays. As shown in Fig. 2E and F, GA could significantly inhibit the copies of viral genomes of EMCV and decrease virus titer at 12, 24, and 36 h post-infection, respectively. And the antiviral effect was most pronounced at 24 h with a dose-effect relationship. Therefore, the subsequent experiments were conducted at 24 h.

Simultaneously, the morphological changes of cells in each group were observed. The results showed that compared with the uninfected group, the infected group exhibited increased cell gaps, cell shrinkage, and patches of cell death, while GA could significantly inhibit the cytopathic effects induced by EMCV (Fig. 2G). In addition, the viral protein VP1 of EMCV is the main protein promoting its proliferation. Therefore, the expression of VP1 was detected by immunofluorescence to further verify the inhibitory effect of GA on EMCV. The results showed that GA could significantly reduce the VP1 level in A549 cells (Fig. 2H), indicating that GA could effectively inhibit the expression of VP1. The above results suggested that GA could significantly inhibit the proliferation of EMCV *in vitro*.

## GA exerts anti-EMCV effect by promoting the expression of IFN-β

After the GA treatment of EMCV-infected HEK-293 cells for 24 h, the expression of Bax, Bcl-2, TNF-α, IL-6, and IFN-β was detected by RT-qPCR and ELISA. Compared with the infected group, treatment with GA (80/160 µM) significantly increased the expression of IFN-β in EMCV-infected HEK-293 cells. However, it had no significant effect on the expressions of Bax, Bcl-2, TNF-α, and IL-6 (Fig. 3A through G). These results indicated that GA might inhibit EMCV replication by increasing the production of IFN-β. And this effect had nothing to do with the drug itself (Fig. 3H). Further studies demonstrated that GA could promote the expression of IFN-β at 12, 24, and 36 h post-infection (Fig. 3I). The above results indicated that GA could exert anti-EMCV effect by promoting the expression of IFN-β.

## GA induces the expression of IFN-β by regulating the RLR signaling pathway

After HEK-293 cells were infected with EMCV, the protein levels of TBK1 and p-IRF3 increased significantly. Compared with the mock group, the expression of TBK1, p-TBK1, IRF3, and p-IRF3 increased significantly following GA intervention (Fig. 4A). As shown in Fig. 4B through D, to further verify the target of GA, HA-MDA5, Flag-TBK1, and Flag-IRF3 (5D) were overexpressed in HEK-293 cells, and the effect of GA on the mRNA of IFN-β was examined. The results demonstrated that after overexpressing TBK1 and IRF3, GA could significantly elevate the mRNA level of IFN-β. The above results suggest that GA may promote the production of IFN-β by activating the RLR signaling pathway and regulating the expression of TBK1 and IRF3.

## GA induces the expression of IFN-β by promoting the interaction between TBK1 and IRF3

HEK-293 cells were transfected with Poly(I:C) to activate the RLR signaling pathway and then treated with GA. The results showed that after Poly(I:C) activated the RLR signaling pathway, the expressions of TBK1, p-TBK1, IRF3, and p-IRF3 were all significantly

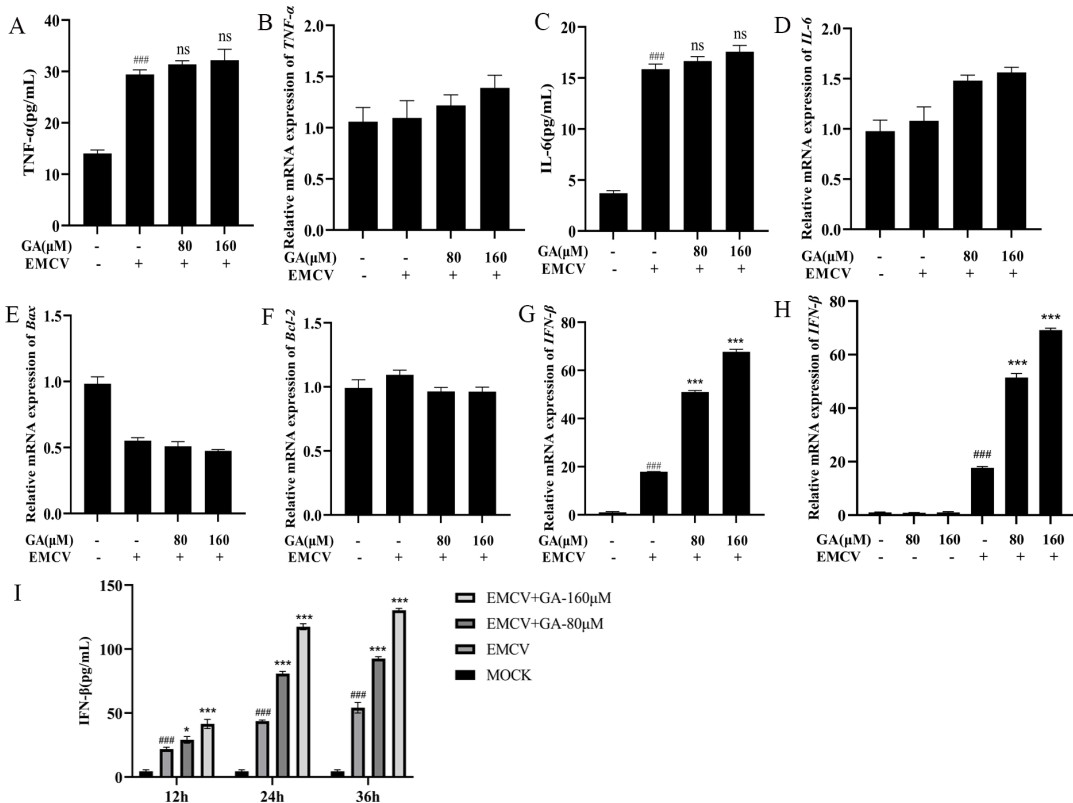

**FIG 3** GA upregulates the expression of IFN-β to inhibit the replication of EMCV. RT-qPCR and ELISA were used to analyze the effects of GA treatment on the expression of TNF-α (A and B), IL-6 (C and D), Bax (E), Bcl-2 (F), and IFN-β (G). (H) In the absence of EMCV infection, the treatment with GA (80/160 µM) did not affect the expression of IFN-β mRNA in HEK-293 cells. The effect of GA on the expression of IFN-β at different time points (12, 24, and 36 h) during EMCV infection (I). Data were presented as the mean ± standard deviation (SD) of three independent experiments, and $P$ values indicate the results of one-way ANOVA with Tukey's multiple comparison test or two-tailed t tests. ***$P < 0.001$, **$P < 0.01$, and *$P < 0.05$ vs EMCV group; ###$P < 0.001$, ##$P < 0.01$, and #$P < 0.05$ vs mock group.

upregulated. Compared with the Poly(I:C) group, GA could further significantly enhance the expressions of TBK1, p-TBK1, IRF3, and p-IRF3 (Fig. 5A) and significantly increase the level of IFN-β mRNA (Fig. 5B). To further explore the mechanism of GA-induced expression of IFN-β, the effect of GA on the interaction between TBK1 and IRF3 was detected using the co-immunoprecipitation method. The results showed that GA could promote the interaction between TBK1 and IRF3 (Fig. 5C). In addition, an RNAi assay was performed in HEK-293. The results showed that the IRF3 (si003) expression was decreased significantly in the siRNA transfection group, as shown in Fig. 5D. Then we analyzed the effect of IRF3 knockdown on GA's anti-EMCV. As shown in Fig. 5E, following GA addition, the virus titers of the IRF3 knockdown HEK-293 cells were higher than those of the siNC control cells. Besides, the expression of IFN-β was decreased in the siIRF3 group compared to the control siNC group (Fig. 5F). The above results suggest that GA targeted IRF3 to regulate the interaction between TBK1 and IRF3 to induce the expression of IFN-β, thereby playing an anti-EMCV role.

## GA alleviates EMCV infection in mice

The LD$_{50}$ of GA was not detected in mice acute toxicity test. According to the reference, the therapeutic doses of GA were set at 40 mg/kg (L-GA) and 80 mg/kg (H-GA), respectively. During the experiment, mice infected with EMCV showed depression, huddling, unkempt and dull fur, reduced dietary wishes, and neurological symptoms such as hind limb paralysis on the fourth day. GA could ameliorate the above symptoms. Moreover, GA significantly increased the survival rate of mice (Fig. 6B) and effectively

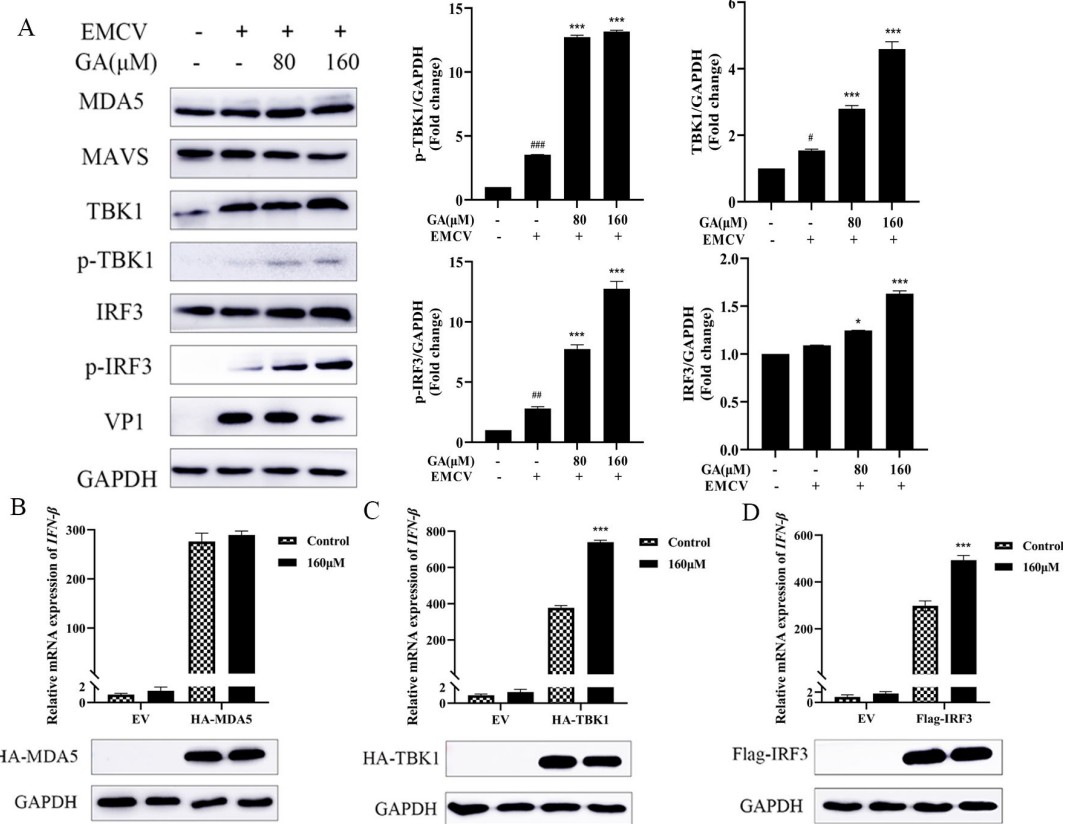

**FIG 4** GA regulates the RLR signaling pathway to inhibit EMCV replication. (A) Compared with the cells treated with 0 µM GA, those treated with GA (80/160 µM) significantly upregulated the expression of TBK1, p-TBK1, IRF3, and p-IRF3. (B–D) The overexpression of TBK1 and IRF3 significantly increased the expression of IFN-β. GAPDH was used as a control. Data were presented as the mean ± standard deviation (SD) of three independent experiments with $P$ values indicating the results of one-way ANOVA and Tukey's multiple comparison test or two-tailed t tests. ***$P$ < 0.001, **$P$ < 0.01, and *$P$ < 0.05 vs EMCV group; ###$P$ < 0.001, ##$P$ < 0.01, and #$P$ < 0.05 vs mock group.

prevented weight loss in mice following EMCV infection (Fig. 6C). To further investigate the protective effects of GA on different organs of EMCV-infected mice, the copies of viral genomes in the brain tissues and heart of mice were detected. As shown in Fig. 6D and E, the results showed that GA could remarkably reduce the copies of viral genomes in the brain tissues and heart of mice. The $TCID_{50}$ results indicated that GA could significantly decrease the viral titer in the brain tissues of mice (Fig. 6F). Simultaneously, GA significantly promoted the production of IFN-β in the brain tissues of EMCV-infected mice (Fig. 6G). The above results suggested that GA exerted a significant antiviral effect in mice infected with EMCV.

## DISCUSSION

As a zoonotic virus, EMCV not only severely impacts the pig husbandry but also presents a potential threat to human health. There is an urgent necessity to develop novel drugs for its treatment. GA has widespread pharmacological effects, including antioxidant, anti-inflammatory, antibacterial, and anti-tumor properties. Studies have demonstrated that it could inhibit a variety of viruses (20–23); however, its specific antiviral mechanism remains unclear. The results of this study indicate that GA has a certain inhibitory effect on EMCV. It can significantly decrease the viral titer of EMCV, effectively reduce cell rupture and intercellular spaces induced by EMCV infection, and display good cytoprotective effects. VP1, as the main structural protein of EMCV, is its major antigenic determinant domain, which is closely associated with the virus pathogenicity. It also

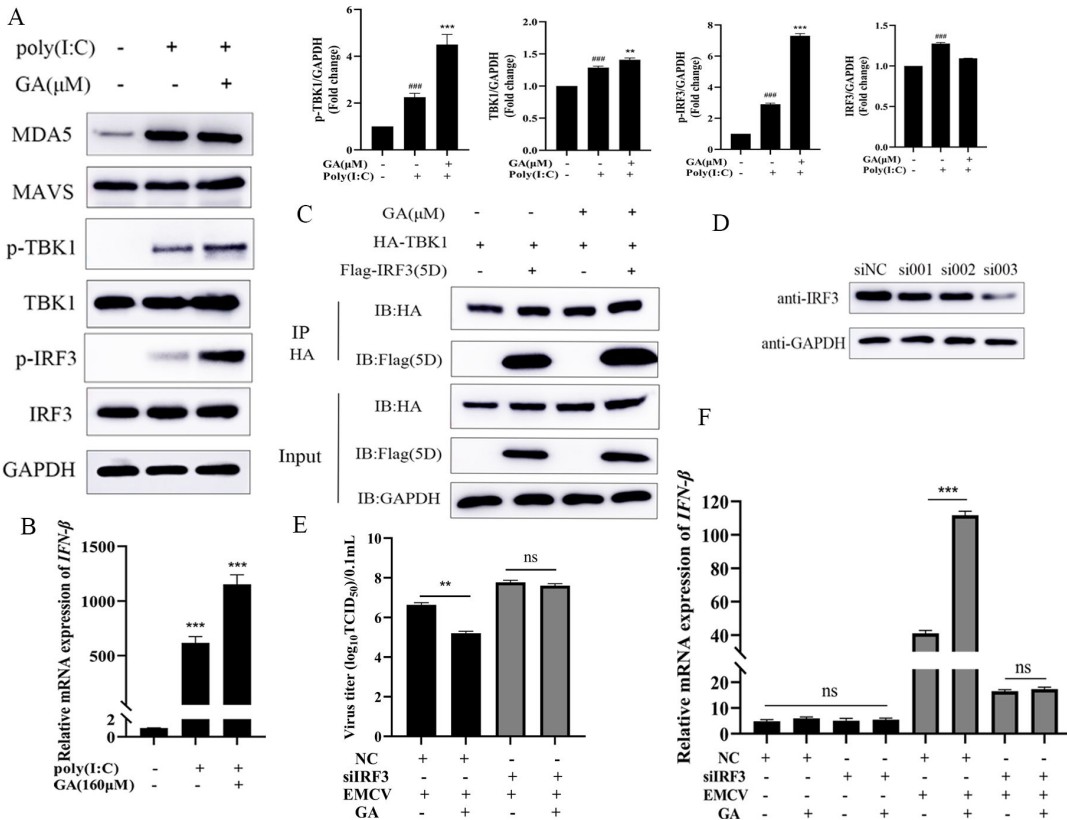

**FIG 5** GA induces the expression of IFN-β by targeting IRF3 to promote the interaction between TBK1 and IRF3. (A) GA (160 µM) significantly upregulated the expressions of TBK1, p-TBK1, IRF3, and p-IRF3. (B) GA (160 µM) significantly upregulated the expression of IFN-β. (C) GA could promote the interaction between TBK1 and IRF3. (D) siRNA targeting IRF3 was determined by Western blotting. After siRNA targeting IRF3, GA had no significant effect on the virus titer (E) and the expression of IFN-β (F) during EMCV infection. Data were presented as the mean ± standard deviation (SD) of three independent experiments, and *P* values indicate the results of one-way ANOVA with Tukey's multiple comparison test or two-tailed t tests. \*\*\**P* < 0.001, \*\**P* < 0.01, and \**P* < 0.05 vs EMCV group; ###*P* < 0.001, ##*P* < 0.01, and #*P* < 0.05 vs mock group.

interacts with receptors on the cell surface and can stimulate the body to generate neutralizing antibodies (2, 24, 25). Our results showed that GA can remarkably reduce the expression of the viral protein VP1 in cells.

In order to further explore the mechanism of GA's antiviral effect, the regulatory effect of GA on TNF-α, IL-6, Bcl2, Bax, and IFN-β was detected. During viral infection, the NF-κB-signaling pathway is activated. Meanwhile, the dimer of p65/p50 is released from the cytoplasm into the nucleus and induces the expression of cytokines, further recruiting immune cells to clear viral infections. In addition, the infected cells promoted apoptosis, thereby inhibiting the replication and spread of the virus (26, 27). Bax, a pro-apoptotic protein within the Bcl2 family, promotes cell apoptosis by increasing the permeability of the mitochondrial membrane and releasing cytochrome. Bcl-2 is an anti-apoptotic protein, mainly distributed on the outer membrane of mitochondria, and it maintains cell survival by inhibiting apoptosis (28). Our results showed that GA had no significant effects on TNF-α, IL-6, Bcl2, and Bax, but it could significantly upregulate the expression of IFN-β during EMCV infection, which revealed that GA could inhibit the replication of EMCV by regulating the expression of IFN-β, rather than the NF-κB signal pathway and cell apoptosis.

IFN-β, as a type I IFN, plays a crucial role in the innate immune system of the body against viral infection. The host innate immunity recognizes pathogen-associated molecular patterns of viruses via pattern recognition receptors. The RLR family consists of RIG-I, melanoma differentiation-associated protein 5 (MDA5), and laboratory genetics

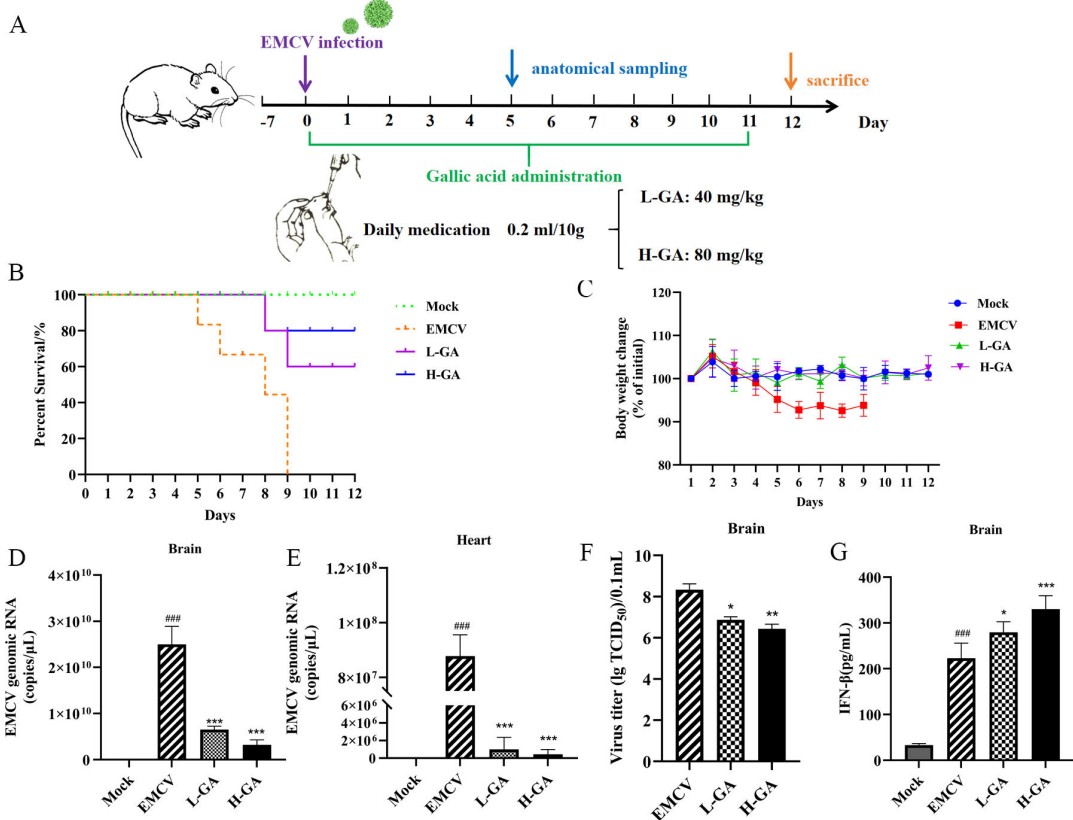

**FIG 6** Protective effect of GA against EMCV infection in the BALB/c mice. (A) Experiment design. Survival curve (B) and body weight change of the mice during the experiment period (C). The copies of viral genomes in brain tissues (D) and heart (E) of the mice. GA significantly decreased virus titer (F) and increased the expression of IFN-β in the brain of mice (G). Data were presented as the mean ± standard deviation (SD) of at least three independent experiments with $P$ values indicating the results of one-way ANOVA and Tukey's multiple comparison test or two-tailed t tests, except for survival curve $n = 8$ per group, the remaining analysis $n = 5$ per group. \*\*\*$P < 0.001$, \*\*$P < 0.01$, and \*$P < 0.05$ vs EMCV group; ###$P < 0.001$, ##$P < 0.01$, and #$P < 0.05$ vs mock group.

and physiology protein 2 (LGP2) (1, 29, 30). Among them, MDA5 mainly recognizes long double-stranded RNA (dsRNA). The dsRNA of the replication intermediate of EMCV is approximately 7.8 kb, which falls into the category of long dsRNA. Furthermore, our previous studies have demonstrated that EMCV-infected cells mainly activate the innate immune antiviral response by recognizing its RNA through MDA5 (30, 31). When MDA5 binds to dsRNA, it will promote its oligomerization and interact with MAVS. The activated MAVS recruits and activates TBK1/IKKε. TBK1/IKKε phosphorylates IRF3/IRF7 through its kinase activity, thereby inducing IFN-I to exert antiviral effects (6, 32–34). Therefore, drugs can effectively control viral infections by regulating the RLR signaling pathway. Existing studies have shown that baicalin could upregulate the expressions of IFN-I and IFN-III and their receptors under the stimulation of Poly (I:C) (35). Consistent with the above results, the findings of this study suggested that GA exerted anti-EMCV effect by promoting the expression of IFN-β through regulating the RLR signaling pathway. Specifically, GA targeted TBK1 to interact with IRF3, leading to a conformational change in phosphorylated IRF3, forming a dimer, translocating from the cytoplasm to the nucleus, binding to its co-activators, and then initiating the production of IFN-β. In summary, GA exerted an antiviral effect against EMCV infection by activating TBK1 and IRF3 in the RLR signaling pathway and promoting the production of IFN-β.

In this study, we further validated the protective effect of GA in the EMCV-infected mouse model. We measured the changes in copies of viral genomes in the brain tissues and hearts, the viral titer, and the level of IFN-β in the brain tissues of EMCV-infected mice. The results indicated that GA could significantly decrease the copies of viral

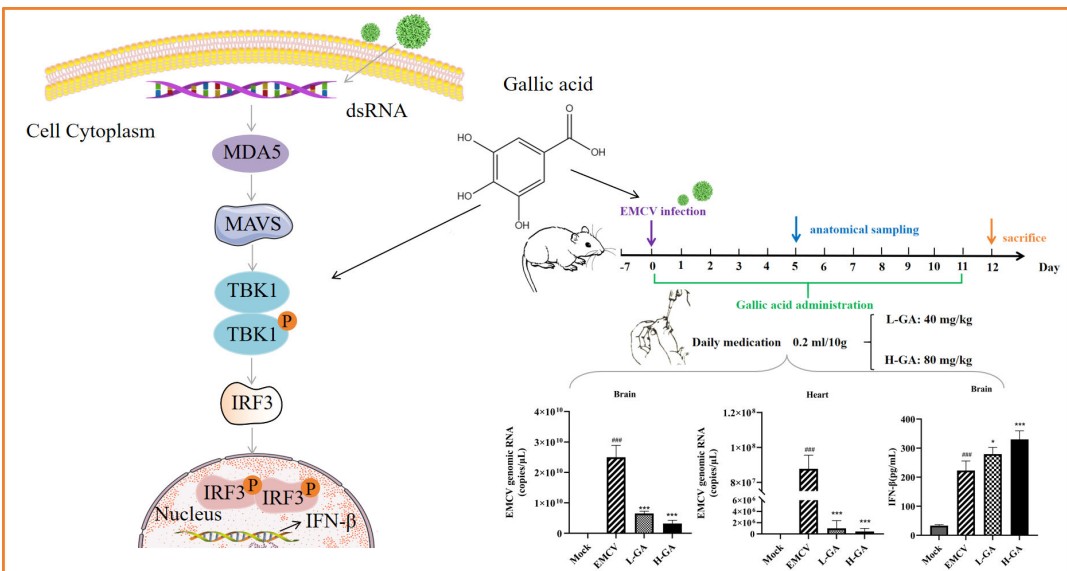

**FIG 7** Schematic diagram showing that GA targets TBK1/IRF3 *in vitro*, promotes the expression of IFN-β, inhibits EMCV, and exhibits a protective effect on mice *in vivo*.

genomes in the brain tissues and hearts of mice, as well as the viral titer in the brain tissue, and significantly enhance the expression of IFN-β in the brain tissues. It was further demonstrated that GA resisted EMCV infection by upregulating the expression of IFN-β.

## Conclusion

This study demonstrates that GA can effectively inhibit the infection of EMCV both *in vitro* and *in vivo*. The specific mechanism of its anti-EMCV mainly involves promoting the interaction between TBK1 and IRF3 and subsequently upregulating the expression of IFN-β to inhibit the proliferation of EMCV. This research provides guidance for the clinical treatment of EMCV-related diseases, offers new insights into elucidating the pharmacodynamic mechanism of GA, and provides a theoretical foundation for expanding its clinical applications and developing drugs against EMCV (Fig. 7).

### ACKNOWLEDGMENTS

This work was supported by Research Center of Traditional Chinese Medicine, Gansu Province (zyzx-2023-22).

Yan Zhang and Yanqiao Wen contributed to the research design, experiment development, data analysis, and article drafting. Weijiao Xue and Tengyu Zhang contributed to animal experiments. Jixia Hou and Xuewen Chen contributed to data analysis. Ruofei Feng and Chunxia Tan contributed equally to the conception, design, and supervision of the study. All authors are accountable for the integrity and accuracy of all data presented in the article.

### AUTHOR AFFILIATIONS

[1]Gansu University of Chinese Medicine, Lanzhou, China
[2]Research Center of Traditional Chinese Medicine, Lanzhou, China
[3]Key Laboratory of Biotechnology and Bioengineering of State Ethnic Affairs Commission, Biomedical Research Center, Northwest Minzu University, Lanzhou, China
[4]School of Life Sciences and Engineering, Northwest Minzu University, Lanzhou, China

## AUTHOR ORCIDs

Yanqiao Wen http://orcid.org/0000-0002-8401-0526

Ruofei Feng http://orcid.org/0000-0001-9647-2827

Chunxia Tan http://orcid.org/0000-0003-4096-2522

## ADDITIONAL FILES

The following material is available online.

## Open Peer Review

**PEER REVIEW HISTORY (review-history.pdf).** An accounting of the reviewer comments and feedback.

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
