## [Reviewer comments · Microbiology Spectrum]

Microbiology Spectrum

Gallic acid inhibits EMCV infection via targeting the interaction between TBK1 and IRF3 to promote IFN- β expression

Yan Zhang, Yanqiao Wen, Weijiao Xue, Tengyu Zhang, Jixia Hou, Xuewen Chen, Chunxia Tan, and Ruofei Feng

Corresponding Author(s): Ruofei Feng, Northwest Minzu University

Review Timeline:

Submission Date:	May 15, 2025
Editorial Decision:	June 9, 2025
Revision Received:	August 9, 2025
Accepted:	August 28, 2025

Editor: Yunyu Chen

Reviewer(s): Disclosure of reviewer identity is with reference to reviewer comments included in decision letter(s). The following individuals involved in review of your submission have agreed to reveal their identity: Huimin Liu (Reviewer #1)

Transaction Report:

DOI: <https://doi.org/10.1128/spectrum.01524-25>

Re: Spectrum01524-25 (**Gallic acid inhibits EMCV infection via targeting the interaction between TBK1 and IRF3 to promote IFN- β expression**)

Dear Prof. Ruofei Feng:

Thank you for the privilege of reviewing your work. Below you will find my comments, instructions from the Spectrum editorial office, and the reviewer comments.

The authors mention that they approved by the Animal Ethics Committee of Northwest Minzu University and the Animal Protection and Utilization Committee. The authors should provide official number for animal permission for this project.

Revision Guidelines

Sincerely,
Yunyu Chen
Editor
Microbiology Spectrum

Reviewer #1 (Comments for the Author):

The manuscript titled "Gallic acid inhibits EMCV infection via targeting the interaction between TBK1 and IRF3 to promote IFN- β expression" investigates the antiviral mechanism of gallic acid (GA) against Encephalomyocarditis virus (EMCV). The study is well-structured and addresses an important gap in the field of antiviral therapeutics. However, several issues need to be

addressed to improve rigor.

Major comments:

1. Should include dose-response and time-course data for GA's antiviral effects: The 24-hour endpoint for most assays may not capture dynamic changes in viral replication or IFN- β induction, Earlier/later time points should be included.
2. The claim that GA selectively upregulates IFN- β (but not TNF- α /IL-6) needs validation. Are these cytokines measured at the protein level (ELISA)? mRNA data alone are insufficient. Strengthen mechanistic claims with direct binding/kinase assays or knockdown experiments.
3. Clarify statistical methods (e.g., n-values for in vivo studies) and figure labels.
4. The conclusion that GA directly targets TBK1-IRF3 is overstated without evidence. Alternative mechanisms (e.g., GA's antioxidant effects on MAVS/TBK1 activation) should be discussed.

Minor comments:

1. Many minor errors exist in the format of the manuscript.

Reviewer #3 (Comments for the Author):

As per the attachment

The manuscript titled "Gallic acid inhibits EMCV infection via targeting the interaction between TBK1 and IRF3 to promote IFN- β expression" investigates the antiviral mechanism of gallic acid (GA) against Encephalomyocarditis virus (EMCV). The study is well-structured and addresses an important gap in the field of antiviral therapeutics. However, several issues need to be addressed to improve rigor.

Major comments:

1. Should include dose-response and time-course data for GA's antiviral effects: The 24-hour endpoint for most assays may not capture dynamic changes in viral replication or IFN- β induction, Earlier/later time points should be included.
2. The claim that GA selectively upregulates IFN- β (but not TNF- α /IL-6) needs validation. Are these cytokines measured at the protein level (ELISA)? mRNA data alone are insufficient. Strengthen mechanistic claims with direct binding/kinase assays or knockdown experiments.
3. Clarify statistical methods (e.g., n-values for in vivo studies) and figure labels.
4. The conclusion that GA directly targets TBK1-IRF3 is overstated without evidence. Alternative mechanisms (e.g., GA's antioxidant effects on MAVS/TBK1 activation) should be discussed.

Minor comments:

1. Many minor errors exist in the format of the manuscript.

Comments

Title: Gallic acid inhibits EMCV infection via targeting the interaction 1 between TBK1 and IRF3 to promote IFN- β expression (control no. Spectrum 01524-25)

The manuscript of Yan Zhang et al. describes the pathway mechanism of gallic acid on antiviral effect (against EMCV) which is not has been studied and published in anywhere and the topic is also interesting. Therefore, I recommend publication in Microbiology Spectrum after the following minor corrections:

1. The author should elaborate in the introduction on the significance of the gene about the mechanisms of viral infection or host cellular pathways and the discussion section should provide a more comprehensive and in-depth interpretation of the findings. for example, bcl-2, BAX.
2. In plasmid transfection, the protocol should mention the sufficient detail about whether you transfect a single plasmid or undergo co-transfection (with plasmid HA-TBK1, Flag-IRF-3)?
3. In a standard virus titer assay, we do not apply any antiviral compound. But in your manuscript, you added GA at two concentrations? Why?
4. Overall, I suggested revising English writing and there are some instances of improper grammar and sentence structure in the context. Furthermore, there are numerous abbreviations throughout contexts. Some abbreviations are mentioned for the first time, there are no full names!
5. Fig 4A, as a graph bar, the figure shows the same symbol (-/-), please check!
6. The caption for Fig. 6 is inconsistent with the figure content. The author is advised to carefully review and correct it. Furthermore, the absence of the mock group in Figure 6F requires clarification.
7. Some content citations do not correspond with the author's references. As example, reference (1) does not appear to EMCV, raising concerns about its appropriateness as a citation in this context. Reference (9) does not support the claim that “IFN-a, IFN-b” is associated with the antivirus. In addition, some of the references such as Ref. (26) are still

incomplete (no issue, no page). Therefore, the author is advised to thoroughly check the accuracy and consistency of all references throughout the manuscript."

8. The image in Figure 2G and 2H appears to be of low resolution; it is recommended to replace it with a higher resolution version.

Authors' Response to the Reviewer Comments

Manuscript #: Spectrum01524-25R1

Title of Paper: Gallic acid inhibits EMCV infection via targeting the interaction between TBK1 and IRF3 to promote IFN- β expression

Dear Editor,

We appreciate the time and efforts by the editor and two referees in reviewing this manuscript. We have modified the manuscript according to the reviewer's comments and the detailed corrections are listed below point by point. We believed that the revised version can meet the journal publication requirements. Revisions in the manuscript are marked with red highlights for additions.

Kind regards,

Ruofei Feng

8/8/2025

Reviewer's Responses to Questions POINT TO POINT

Reviewer#1 (Comments for the Author):

The manuscript titled "Gallic acid inhibits EMCV infection via targeting the interaction between TBK1 and IRF3 to promote IFN- β expression" investigates the antiviral mechanism of gallic acid (GA) against Encephalomyocarditis virus (EMCV). The study is well-structured and addresses an important gap in the field of antiviral therapeutics. However, several issues need to be addressed to improve rigor.

Major comments:

1. Should include dose-response and time-course data for GA's antiviral effects: The 24-hour endpoint for most assays may not capture dynamic changes in viral replication or IFN- β induction, Earlier/later time points should be included.

We greatly appreciate the reviewer's efforts to carefully review the paper and the valuable suggestions offered. We have supplemented dose-response and time-course data for GA's antiviral effects and IFN- β induction through qPCR and ELISA assays in HEK-293 cells and annotated the changes in red (lines 275-281, lines 293-301). The result demonstrated that GA could significantly inhibit the copies of viral

genomes of EMCV and increase the expression of IFN- β at 12h, 24h and 36h post-infection respectively with a dose-effect relationship. (Corresponding to Fig.2E and 3I in the text).

2. The claim that GA selectively upregulates IFN- β (but not TNF- α /IL-6) needs validation. Are these cytokines measured at the protein level (ELISA)? mRNA data alone are insufficient. Strengthen mechanistic claims with direct binding/kinase assays or knockdown experiments.

Special thanks to the Reviewer for valuable feedback. In the revised manuscript, we have included additional the protein expression level of TNF- α and IL-6 by ELISA assays. We have made modifications marked in red text (in lines 293-301). Our findings revealed that GA had no significant effect on the expression of TNF- α and IL-6, which were consistent with mRNA data of them (Corresponding to Fig.3A and 3C in the text), the results were shown in the specific figure below.

In addition, to further verify mechanistic of GA on anti-EMCV, an RNAi assay was performed in HEK-293 cell. Results showed that IRF3 expression was decreased significantly in the siRNA transfection groups. Then we analyzed the effect of IRF3 knockdown on GA anti-EMCV and IFN- β expression. As shown in Fig.5D-F, when transfecting siRNA following with GA, the virus titers of the IRF3 knockdown was higher than siNC, meanwhile, the expression of IFN- β decreased. Together, these data demonstrated that GA might target IRF3 to regulate TBK1-IRF3, and subsequently to up-regulate the expression of IFN- β to exert anti-EMCV effect. We have made modifications marked in red text (in lines 324-332)

3. Clarify statistical methods (e.g., n-values for in vivo studies) and figure labels.

We much appreciate the reviewer's careful review. In the revised manuscript, the statistical methods were clarified and the n-values were stated in figure labels. The modifications were marked in red.

4. The conclusion that GA directly targets TBK1-IRF3 is overstated without evidence. Alternative mechanisms (e.g., GA's antioxidant effects on MAVS/TBK1 activation) should be discussed.

Thank you for the comprehensive review and detailed suggestions. We have knocked down the downstream factor IRF3 to further verify mechanistic of GA on anti-EMCV. The results showed that IRF3 knockdown in HEK-293 cell could weaken the anti-EMCV effect of GA. Therefore, regulation of TBK1-IRF3 is one of the effective ways for GA to anti-EMCV. Whereas GA's antioxidant effects on MAVS/TBK1 activation is the subject of our further research.

Reviewer#3 (Comments for the Author):

Title: Gallic acid inhibits EMCV infection via targeting the interaction 1 between TBK1 and IRF3 to promote IFN-β expression (control no. Spectrum 01524-25). The manuscript of Yan Zhang et al. describes the pathway mechanism of gallic acid on antiviral effect (against EMCV) which is not has been studied and published in

anywhere and the topic is also interesting. Therefore, I recommend publication in *Microbiology Spectrum* after the following minor corrections:

1. The author should elaborate in the introduction on the significance of the gene about the mechanisms of viral infection or host cellular pathways and the discussion section should provide a more comprehensive and in-depth interpretation of the findings. for example, bcl-2, BAX.

Thanks for pointing out this issue. We have elaborated the antiviral mechanism of IFN- β , TNF- α , IL-6, Bcl2 and Bax in the introduction and discussion section respectively and annotated the changes in red (lines 66-73 and 363-376).

2. In plasmid transfection, the protocol should mention the sufficient detail about whether you transfect a single plasmid or undergo co-transfection (with plasmid HA-TBK1, FlagIRF-3)?

We sincerely appreciate the constructive feedback. In plasmid transfection, the plasmid HA-TBK1 and FlagIRF-3 were undergone co-transfection during the co-immunoprecipitation assay, while all the others transfected single plasmid. We have made modifications and annotated the changes in red (lines 189-191).

3. In a standard virus titer assay, we do not apply any antiviral compound. But in your manuscript, you added GA at two concentrations? Why?

Thanks for your careful evaluation of manuscript. The samples used for virus titer detection were collected with or without GA treatment. It didn't mean when we did virus titer assay followed with GA addition. The aim of this method was to verify the antiviral effect of GA during EMCV replication in HEK-293 cells.

4. Overall, I suggested revising English writing and there are some instances of improper grammar and sentence structure in the context. Furthermore, there are numerous abbreviations throughout contexts. Some abbreviations are mentioned for the first time, there are no full names!

Thank you for your suggestion. We have carefully revised English writing, including grammar and sentence structure in the context and annotated the changes in red.

5. Fig 4A, as a graph bar, the figure shows the same symbol (-/-), please check!

Thanks for your careful evaluation of manuscript. According to your suggestion, we have made modified them to the same symbol in Fig. 4A.

6. The caption for Fig. 6 is inconsistent with the figure content. The author is advised to carefully review and correct it. Furthermore, the absence of the mock group in Figure 6F requires clarification.

We much appreciate the reviewer's careful review. We have rechecked and made modifications marked in the red the caption for Fig. 6. In addition, the mice of mock group weren't challenged with EMCV, which the virus titer was undetected. Therefore, the data was not included in Fig.6F when dealt with the virus titers in the other experimental groups.

7. Some content citations do not correspond with the author's references. As example, reference (1) does not appear to EMCV, raising concerns about its appropriateness as a citation in this context. Reference (9) does not support the claim that "IFN-a, IFN-b" is associated with the antivirus. In addition, some of the references such as Ref. (26) are still incomplete (no issue, no page). Therefore, the author is advised to thoroughly check the accuracy and consistency of all references throughout the manuscript."

We greatly appreciate the reviewer's efforts to carefully review the paper. In the revised manuscript, we have rechecked and replaced some of references.

8. The image in Figure 2G and 2H appears to be of low resolution; it is recommended to replace it with a higher resolution version.

Thank you for your suggestion. In the updated version, we replaced higher resolution images in Fig. 2G and 2H.

Re: Spectrum01524-25R1 (**Gallic acid inhibits EMCV infection via targeting the interaction between TBK1 and IRF3 to promote IFN- β expression**)

Dear Prof. Ruofei Feng:

Your manuscript has now been reviewed by expert reviewers for the journal. Based on the reviewer's comments, as well as my own review, we would be happy to accept your manuscript in its current form for publication in Microbiology Spectrum.

Now, I am forwarding it to the ASM production staff for publication. Your paper will first be checked to make sure all elements meet the technical requirements. ASM staff will contact you if anything needs to be revised before copyediting and production can begin. Otherwise, you will be notified when your proofs are ready to be viewed.

Sincerely,
Yunyu Chen
Editor
Microbiology Spectrum

Dear editor,

Thank you for sending the revised manuscript. I am satisfied by the changes have made in response to my comments. They have addressed all my concerns satisfactorily, and the manuscript has been significantly improved. I believe it is now suitable for publication in *Microbiology Spectrum*.

Best wishes

Huimin Liu